

# Bilateral photon emission from a vibrating mirror and multiphoton entanglement generation

Alberto Mercurio[1,2,3⋆], Enrico Russo[4,5,6†], Fabio Mauceri[1], Salvatore Savasta[1,4], Franco Nori[4,7,8], Vincenzo Macrì[4,5,9‡] and Rosario Lo Franco[5]

**1** Dipartimento di Scienze Matematiche e Informatiche, Scienze Fisiche e Scienze della Terra, Università di Messina, I-98166 Messina, Italy
**2** Laboratory of Theoretical Physics of Nanosystems (LTPN), Institute of Physics, Ecole Polytechnique Fédérale de Lausanne (EPFL), CH-1015 Lausanne, Switzerland
**3** Center for Quantum Science and Engineering, EPFL, CH-1015 Lausanne, Switzerland
**4** Theoretical Quantum Physics Laboratory, RIKEN, Wako-shi, Saitama 351-0198, Japan
**5** Dipartimento di Ingegneria, Università degli Studi di Palermo, Viale delle Scienze, 90128 Palermo, Italy
**6** University San Pablo-CEU, CEU Universities, Department of Applied Mathematics and Data Science, Campus de Moncloa, C/Juliá n Romea 23 28003, Madrid, Spain
**7** RIKEN Center for Quantum Computing (RQC), Wako-shi, Saitama 351-0198, Japan
**8** Physics Department, The University of Michigan, Ann Arbor, Michigan 48109-1040, USA
**9** Departamento de Física Teórica de la Materia Condensada and Condensed Matter Physics Center (IFIMAC), Universidad Autónoma de Madrid, E-28049 Madrid, Spain

⋆ alberto.mercurio96@gmail.com , † enrico.russo@a.riken.jp , ‡ macrivince1978@gmail.com

## Abstract

**Entanglement plays a crucial role in the development of quantum-enabled devices. One significant objective is the deterministic creation and distribution of entangled states, achieved, for example, through a mechanical oscillator interacting with confined electromagnetic fields. In this study, we explore a cavity resonator containing a two-sided perfect mirror. Although the mirror separates the cavity modes into two independent confined electromagnetic fields, the radiation pressure interaction gives rise to high-order effective interactions across all subsystems. Depending on the chosen resonant conditions, which are also related to the position of the mirror, we study $2n$-photon entanglement generation and bilateral photon pair emission. Demonstrating the non-classical nature of the mechanical oscillator, we provide a pathway to control these phenomena, opening potential applications in quantum technologies. Looking ahead, similar integrated devices could be used to entangle subsystems across vastly different energy scales, such as microwave and optical photons.**

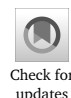

# 1 Introduction

The ability to control quantum mechanical systems using radiation pressure has given rise to the field of optomechanics [1–4], an interesting platform for exploring the quantum properties of mesoscopic objects [5–7]. Cavity optomechanical systems, which involve the interaction between mechanical vibrations and electromagnetic fields, hold the potential for observing quantized vibrational modes in macroscopic objects, even reaching their ground state [8–15]. This opens the door to creating entangled and superposition macroscopic states, paving the way for novel approaches to processing and storing quantum information [16–20].

In general, when electromagnetic quantum fluctuations interact with a very fast-oscillating boundary condition, pairwise real excitations can be created from the vacuum of the electromagnetic field [21–23]. Such a purely quantum phenomenon is known as the dynamical Casimir effect (DCE) [24, 25], which has been experimentally realized in superconducting circuits [26] and Josephson metamaterials [27].

Cavity optomechanics involves the modulation of boundary conditions through a mobile mirror, enabling the observation of the DCE. In this scenario, the fundamental process involves the conversion of mechanical energy into photons [28]. A detailed derivation of the optomechanical Hamiltonian can be found in Ref. [29]. Subsequent advancements extended this model to incorporate incoherent excitation of the mirror [30, 31], and other works examined back-reaction and dissipation effects within this framework [32, 33]. Notably, investigations have expanded to consider a cavity with two mobile mirrors [34–36]. In this case, the cavity field facilitates an effective interaction between the two mirrors, resulting in phonon hopping. This broader exploration adds depth to our understanding of the complex dynamics within optomechanical systems.

The present work investigates a cavity resonator equipped with a two-sided perfect mirror embedded within. This configuration corresponds to a tripartite system, where two separated electromagnetic fields interact with the vibrating mirror by radiation pressure (see Fig. 1). Despite the mirror separating the cavity modes into two distinct electromagnetic fields, the radiation pressure interaction induces high-order processes across all subsystems. Recently,

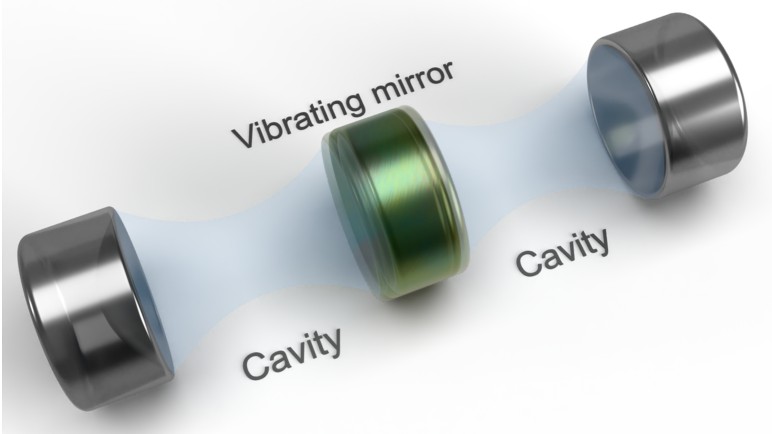

Figure 1: Pictorial representation of our setup. Two electromagnetic cavities separated by a movable two-sided perfect mirror.

this configuration has been studied, shedding light on the dressed ground state and the correlation between the two cavity modes [37], and on the optomechanically induced two-photon hopping effect [38].

Furthermore, path-entangled microwave radiation was observed from *strongly driven* microwave resonators [39], where the entanglement of two distinct driven resonators is generated thanks to the presence of a common mechanical membrane. Here, instead, we propose a scheme to generate $2n$-photon entanglement (e.g., two-, four-photon) and bilateral photon pair emission (that we name the Janus effect), already in an *undriven* setup, which can be achieved with few photonic excitations (allowing us to explore more easily the quantum properties of the system). In particular, the emerging entangled states have the structure of NOON states, which are important in quantum metrology and quantum sensing for their ability to allow precision phase measurements [40]. As a process involving only a few photons, this setup facilitates the examination of the quantum properties of the states. Additionally, it offers enhanced resilience to losses, which increase with the number of photons. The measurement of quantum correlations between the two cavities can be seen as direct evidence of the quantum nature of mesoscopic mechanical objects, without measuring them directly [41, 42].

Each process is activated by a specific resonance condition, which depends on the resonance frequency of the three subsystems, and thus on the position of the mirror. Throughout our analysis, we carry out analytical aspects and numerical simulations to delve into the resonant dynamics and the interplay between the system's parameters, including coupling strengths, bare frequencies, and initial conditions.

In principle, the effects predicted in this work could be experimentally observed using circuit optomechanical systems, namely, employing mechanical micro- or nano-resonators operating in the ultra-high-frequency range within the GHz spectral domain [43, 44]. Despite the current limitations of the experimental feasibility of reaching these resonance conditions, the technology behind the optomechanical systems is advancing very fast. With this theoretical proposal we hope to stimulate future experimental realizations. Moreover, the addition of artificial atoms in a superconducting microwave setup strengthens the coupling with the mechanical resonator [45–49], making it a very promising setup. A valuable alternative approach would entail employing a quantum simulator [50, 51], wherein two LC circuits emulate the cavities, and a superconducting quantum interference device (SQUID) takes on the role of the high-frequency vibrating mirror.

The article is structured as follows: in Section 2 we introduce our quantum model, analyzing in detail three specific resonance conditions: (i) two-photon entanglement generation in Section 2.1; (ii) four-photon entanglement generation in Section 2.2; (iii) bilateral photon emission, which we call Janus effect in Section 2.3. Finally, in Section 3 we offer concluding remarks and outline potential trails for future research in this field. Some details are left on the Appendices. More precisely, in Appendix A we employ the Schrieffer-Wolff method to derive the effective Hamiltonians, and we also show all the coefficients related to the Janus effective Hamiltonian. In Appendix B, we study the convergence of the quantum trajectories to the master equation, when averaging over different number of quantum trajectories. Finally, in Appendix C we study the dependence of the entanglement generation as a function of the detuning.

## 2  Quantum model

Consider two non-interacting single-mode cavities separated by a vibrating two-sided perfect mirror, as sketched in Fig. 1. The three bosons are described by ladder operators $\hat{a}(\hat{a}^\dagger), \hat{c}(\hat{c}^\dagger)$ for the cavities and $\hat{b}(\hat{b}^\dagger)$ for the mirror, satisfying canonical commutation relations. The Hamiltonian can be derived by quantizing the classical Lagrangian description (see Appendix in Ref. [38]), and it reads ($\hbar = 1$)

$$\hat{H} = \omega_a \hat{a}^\dagger \hat{a} + \omega_b \hat{b}^\dagger \hat{b} + \omega_c \hat{c}^\dagger \hat{c} - \frac{g}{2}\big[(\hat{a} + \hat{a}^\dagger)^2 - \Omega^2(\hat{c} + \hat{c}^\dagger)^2\big](\hat{b} + \hat{b}^\dagger), \tag{1}$$

where $\omega_a, \omega_c$ are the bare frequencies of the cavities, $\omega_b$ is the bare frequency of the mirror, $g$ is the coupling strength, and the ratio $\Omega = \omega_c/\omega_a$ is related to the mirror position. In the limit of large detuning $\omega_b \ll \omega_{a,c}$, the rotating wave approximation can be applied, and the standard optomechanical interaction term, proportional to the number operators $\hat{a}^\dagger\hat{a}(\hat{c}^\dagger\hat{c})$, is obtained [3]. It is worth noting the presence of the minus sign in front of the $\Omega$ factor. This arises from the given configuration, where the radiation pressure of the right cavity pushes the mirror in the opposite direction with respect to the radiation pressure of the left cavity [38].

With this Hamiltonian, we will describe three peculiar configurations. By employing the Schrieffer-Wolff approach (see Refs. [52–54] and Appendix A), we obtain effective Hamiltonians which directly show the high-order non-linear processes related to specific resonance conditions.

Although we analytically characterize all these processes by using effective Hamiltonians, all the simulations are carried out by employing the quantum trajectory approach [55, 56], and using the full Hamiltonian in Eq. (1). To explore the phenomenology, we numerically calculate the time evolution of the mean values of the *dressed* number operators [57, 58], i.e., $\langle \hat{\mathcal{X}}_o^- \hat{\mathcal{X}}_o^+ \rangle$ ($o \in \{a, b, c\}$). Each dressed operator is defined as

$$\hat{\mathcal{X}}_o^+ \equiv \sum_{j>k} \langle k|(\hat{o} + \hat{o}^\dagger)|j\rangle\,|k\rangle\langle j|, \quad \hat{\mathcal{X}}_o^- = (\hat{\mathcal{X}}_o^+)^\dagger, \tag{2}$$

where $|j\rangle$ is the $j$-th eigenstate of the full Hamiltonian in Eq. (1). This properly defines the jump operators, which by construction act like an annihilation (creation) operator in the energy basis. In this dressed picture, the quantum jumps are between the dressed states (the eigenstates of the full Hamiltonian) which contain contributions from bare states with an arbitrary number of excitations. With this notation, we refer to the mean value of the number operator of the single quantum trajectory, while average quantities obtained over 1000 quantum trajectories are indicated as $\overline{\langle \hat{\mathcal{X}}_o^- \hat{\mathcal{X}}_o^+ \rangle}$. The dissipation rates of the three subsystems are indicated as $\gamma_a$, $\gamma_b$, and $\gamma_c$.

## 2.1 Two-photon entanglement generation

Let us consider the condition $\omega_a \simeq \omega_b \simeq \omega_c$. The effective Hamiltonian up to the second order becomes (see Appendix A)

$$\hat{H}_{\text{eff}} = \hat{H}_0 + \hat{H}_{\text{shift}} + \hat{H}_{\text{hop}} + \hat{H}_{\text{ent}} + \mathcal{O}(g^3), \tag{3}$$

$$\hat{H}_{\text{shift}} = -\frac{g^2}{3\omega_b} - \frac{4g^2}{3\omega_b}\left(\hat{a}^\dagger\hat{a} + \hat{c}^\dagger\hat{c}\right) - \frac{4g^2}{3\omega_b}\left(\hat{a}^\dagger\hat{a} + 1 + \hat{c}^\dagger\hat{c}\right)\hat{b}^\dagger\hat{b}$$

$$- \frac{5g^2}{6\omega_b}\left(\hat{a}^{\dagger 2}\hat{a}^2 + \hat{c}^{\dagger 2}\hat{c}^2\right) + \frac{2g^2}{\omega_b}\hat{a}^\dagger\hat{a}\hat{c}^\dagger\hat{c},$$

$$\hat{H}_{\text{hop}} = -\frac{g^2}{6\omega_b}\left(\hat{a}^{\dagger 2}\hat{c}^2 + \hat{c}^{\dagger 2}\hat{a}^2\right),$$

$$\hat{H}_{\text{ent}} = -\frac{g^2}{\omega_b}\left((\hat{a}^{\dagger 2} + \hat{c}^{\dagger 2})\hat{b}^2 + (\hat{a}^2 + \hat{c}^2)\hat{b}^{\dagger 2}\right),$$

where $\hat{H}_0 = \omega_a \hat{a}^\dagger\hat{a} + \omega_b \hat{b}^\dagger\hat{b} + \omega_c \hat{c}^\dagger\hat{c}$ is the same in any derivation of effective Hamiltonian.

The Hamiltonian term $\hat{H}_{\text{shift}}$ contains only numbers operators (and their powers) describing bare energy shift due to the perturbation. The two effective interaction terms clarify an otherwise complex dynamic implicit in Eq. (1). Indeed, $\hat{H}_{\text{hop}}$ shows the two-photon hopping sub-process [38] while $\hat{H}_{\text{ent}}$ links the three sub-parts together ultimately bringing to tripartite entanglement.

The effective Hamiltonian in Eq. (3) admits the lowest energy closed dynamics in the sub-Hilbert space spanned by the states $\{|2,0,0\rangle, |0,2,0\rangle, |0,0,2\rangle\}$. Here, the first and third entries represent the $a$ and $c$ cavity excitations, respectively, while the second represents the mirror ($b$) excitation. Under the resonance condition $\omega_a = \omega_c = \omega$, we can define the ordered basis $\{|0,2,0\rangle, |\psi_+^{(2e)}\rangle, |\psi_-^{(2e)}\rangle\}$, with $|\psi_\pm^{(2e)}\rangle = (|2,0,0\rangle \pm |0,0,2\rangle)/\sqrt{2}$ being the symmetric and anti-symmetric two-photon maximally entangled states between the two cavities. Notice that these entangled states have the structure of NOON states, which are typically exploited in quantum-enhanced metrology [40]. Therefore, the Hamiltonian takes the block form

$$H = \begin{pmatrix} 2\omega_b - \frac{3g^2}{\omega_b} & -\frac{2\sqrt{2}g^2}{\omega_b} & 0 \\ -\frac{2\sqrt{2}g^2}{\omega_b} & 2\omega - \frac{5g^2}{\omega_b} & 0 \\ 0 & 0 & 2\omega - \frac{13g^2}{3\omega_b} \end{pmatrix}, \tag{4}$$

highlighting the fact that the dynamics occurs between the state $|0,2,0\rangle$ and $|\psi_+^{(2e)}\rangle$. In other words, two phonons are exchanged with a symmetric entangled state of two totally delocalized photons. The non-uniformity of the matrix elements in Eq. (4) stems from the different coefficients appearing respectively in $\hat{H}_{\text{hop}}$ and $\hat{H}_{\text{ent}}$ as well as the different coefficients appearing in $\hat{H}_{\text{shift}}$. Indeed, in general, the eigenstates of the effective Hamiltonian are written as a generic superposition of $|0,2,0\rangle$ and $|\psi_+^{(2e)}\rangle$. However, we can make the superposition symmetric by choosing the condition $\omega = \omega_b + g^2/\omega_b$, which makes the upper-left block a symmetric matrix with equal diagonal terms. The same condition can be found by minimizing the difference of the two eigenvalues in Eq. (4) as a function of $\omega$. Under this condition the eigenstates of the upper-left block become

$$|\phi_\pm^{(2e)}\rangle = (|0,2,0\rangle \pm |\psi_+^{(2e)}\rangle)/\sqrt{2}. \tag{5}$$

In Fig. 2(a-b) we show two different trajectories, while Fig. 2(c) shows the master-equation-like behavior that arises taking the average over 1000 trajectories. A study on the convergence of the quantum trajectories to the master equation is showed in Appendix B. Both



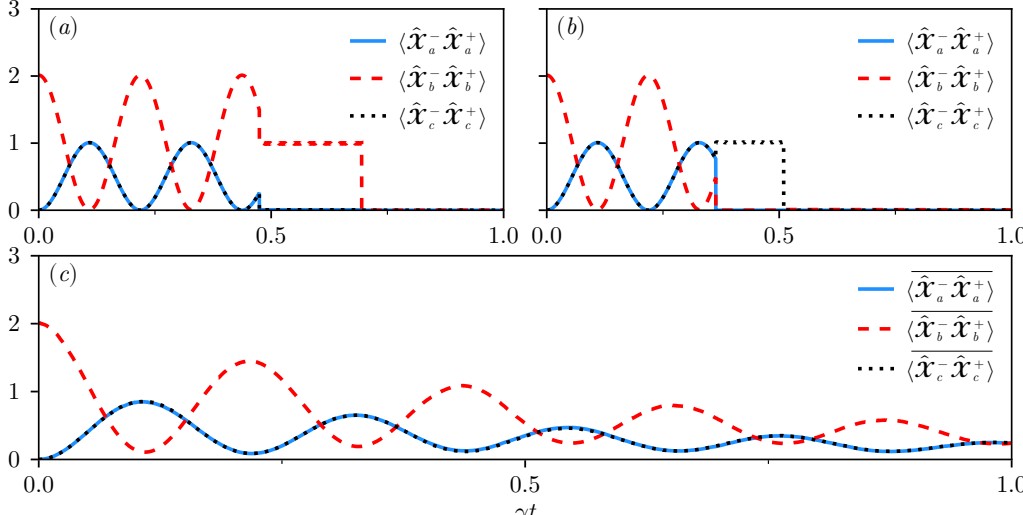

Figure 2: **Two-photon entanglement generation.** The time evolution of the mean values of the dressed number operators for different trajectories and the average over 1000 trajectories. (a) A trajectory where the first jump occurs with a phonon loss, locking the system into the state $|0,1,0\rangle$, until a second quantum jump occurs. (b) A trajectory where the first jump occurs in cavity $c$, locking the system into the state $|0,0,1\rangle$, as a clear signature of an entangled state. (c) The average behavior shows the coherent and dissipative energy exchange between the bare state $|0,2,0\rangle$ and the entangled state $|\psi_+\rangle$. The used parameters are: $\omega_a = \omega_c = \omega_b + g^2/\omega_b$, $g = 0.05\omega_b$, and $\gamma_a = \gamma_b = \gamma_c = 5 \times 10^{-4}\omega_b$.

the snapshots highlight a fascinating trapping effect that occurs whenever one photon is detected in one of the cavities, or when a phonon is detected in the mirror. For instance, in Fig. 2(a) we clearly see that the first jump occurs with a phonon loss, locking the system to the state $|0,1,0\rangle$, while the coherent dynamics is lost. After a certain time, a second jump occurs leaving the system in its ground state. On the contrary, in Fig. 2(b) the first jump occurs in cavity $c$, and the system is locked in the state $|0,0,1\rangle$. Note that, when the cavity $c$ jumps, the number of photons in cavity $a$ immediately goes to zero, as a clear signature of the quantum correlations exhibited in the entangled state $|\psi_+^{(2e)}\rangle$. Again, when the second jump occurs, the system reaches its ground state.

This trapping effect occurs because the effective Hamiltonian does not contain terms that allow the exchange of a single photon-phonon excitation, while the act of measuring (losses) is modeled as a single boson detection. The remaining excitation is localized in the subsystem where the first measurement occurred until a second measurement occurs. In Fig. 2(c) (obtained by averaging over 1000 trajectories) one clearly sees the coherent and dissipative energy exchanging over time, between the bare state $|0,2,0\rangle$ and $|\psi_+^{(2e)}\rangle$, meaning that two-phonon generates two-photon entanglement. Although we primary focus on the generation of two-photon entangled states, the dynamics also exhibit phonon-photon entanglement due to the hybridization of the $|0,2,0\rangle$ and $|\psi_+^{(2e)}\rangle$ states during the Rabi oscillations. This behavior is evident in Fig. 2(a), where the photon number in both cavities drops to zero whenever a jump occurs in the mirror, providing a clear indication of multipartite entanglement. This happens because, after the jump, the system transitions to a manifold with insufficient energy to restore the multipartite entanglement. Note that, the above-mentioned trapping effects are washed out by the averaging of a master equation [59,60]. The parameters used to reproduce Fig. 2 are: $\omega_a = \omega_c = \omega_b + g^2/\omega_b$, $g = 0.05\omega_b$, and $\gamma_a = \gamma_b = \gamma_c = 5 \times 10^{-4}\omega_b$.

It is worth mentioning that, although the two-photon entangled state generation is spontaneously achieved without the presence of a drive, the system initialization requires some pulse sequence to promote the ground state to the desired state. For example, the $|0, 2, 0\rangle$ state can be obtained by using standard procedures [19, 61–64].

## 2.2 Four-photon entanglement generation

Under the resonant conditions, $\omega_b \simeq 4\omega$ ($\omega = \omega_a = \omega_c$) the effective Hamiltonian up to the third order in the coupling constant becomes (see Appendix A)

$$
\begin{aligned}
\hat{H}_{\text{eff}} =& \hat{H}_0 + \hat{H}_{\text{shift}} + \hat{H}_{\text{hop}} + \hat{H}_{\text{ent}}, \\
\hat{H}_{\text{shift}} =& -\frac{2g^2}{3\omega_b} - \frac{5g^2}{3\omega_b}\left(\hat{a}^\dagger\hat{a} + \hat{c}^\dagger\hat{c} + \hat{a}^{\dagger 2}\hat{a}^2 + \hat{c}^{\dagger 2}\hat{c}^2\right) \\
& + \frac{4g^2}{3\omega_b}\left(\hat{a}^\dagger\hat{a} + \hat{c}^\dagger\hat{c} + 1\right)\hat{b}^\dagger\hat{b} + \frac{2g^2}{\omega_b}\hat{a}^\dagger\hat{a}\hat{c}^\dagger\hat{c}, \\
\hat{H}_{\text{hop}} =& \frac{2g^2}{3\omega_b}\left(\hat{a}^{\dagger 2}\hat{c}^2 + \hat{a}^2\hat{c}^{\dagger 2}\right), \\
\hat{H}_{\text{ent}} =& \frac{2g^3}{3\omega_b^2}\left[(\hat{a}^4 - \hat{c}^4)\hat{b}^\dagger + (\hat{a}^{\dagger 4} - \hat{c}^{\dagger 4})\hat{b}\right].
\end{aligned}
\tag{6}
$$

As we did in the previous case we can look for the simplest closed dynamics and project the Hamiltonian in corresponding basis. This subspace is spanned by the states $\{|0, 1, 0\rangle, |2, 0, 2\rangle, |4, 0, 0\rangle, |0, 0, 4\rangle\}$, but due to the form of interaction Hamiltonian part and resonant conditions one can define the ordered basis $\{|0, 1, 0\rangle, |\psi_-^{(4e)}\rangle, |\psi_+^{(4e)}\rangle, |2, 0, 2\rangle\}$, where $|\psi_\pm^{(4e)}\rangle = (|4, 0, 0\rangle \pm |0, 0, 4\rangle)/\sqrt{2}$ being the symmetric and anti-symmetric four-photon maximally entangled states between the two cavities. Again, we highlight that these states have the structure of NOON states. Using this basis, the Hamiltonian takes the block form

$$
H = \begin{pmatrix}
\omega_b + \frac{4g^2}{3\omega_b} & \frac{8g^3}{\sqrt{3}\omega_b^2} & 0 & 0 \\
\frac{8g^3}{\sqrt{3}\omega_b^2} & 4\omega - \frac{80g^2}{3\omega_b} & 0 & 0 \\
0 & 0 & 4\omega - \frac{80g^2}{3\omega_b} & \frac{8g^2}{\sqrt{3}\omega_b} \\
0 & 0 & \frac{8g^2}{\sqrt{3}\omega_b} & 4\omega - \frac{16g^2}{3\omega_b}
\end{pmatrix},
\tag{7}
$$

revealing the two concurrent dynamics: the oscillation between $|0, 1, 0\rangle$ and $|\psi_-^{(4e)}\rangle$ and the one between $|2, 0, 2\rangle$ and $|\psi_+^{(4e)}\rangle$. The latter, namely, the lower-right part of the matrix describing the dynamics between the states $|2, 0, 2\rangle$ and $|\psi_+^{(4e)}\rangle$, can be explained in terms of the two-photon hopping terms [38]. It originates only from the third line in (6), and thus it does not introduce any new effect. Since the mirror plays a role only in the first sub-dynamics, only the upper-left block becomes important for the four-photon entanglement generation process we are describing. To look for symmetric eigenstates, we can proceed as in Section 2.1, by choosing $\omega$ such that the diagonal terms are equal. Note that, in this case, the same cannot be done for the lower-right part because the non-linear $H_{\text{shift}}$ acts differently on these states, making the two dynamics mutually exclusive.

In particular, for $\omega = \frac{\omega_b}{4} + 7\frac{g^2}{\omega_b}$, the eigenstates take the form

$$
|\phi_\pm^{(4e)}\rangle = \frac{1}{\sqrt{2}}\left(|0, 1, 0\rangle \pm |\psi_-^{(4e)}\rangle\right),
\tag{8}
$$

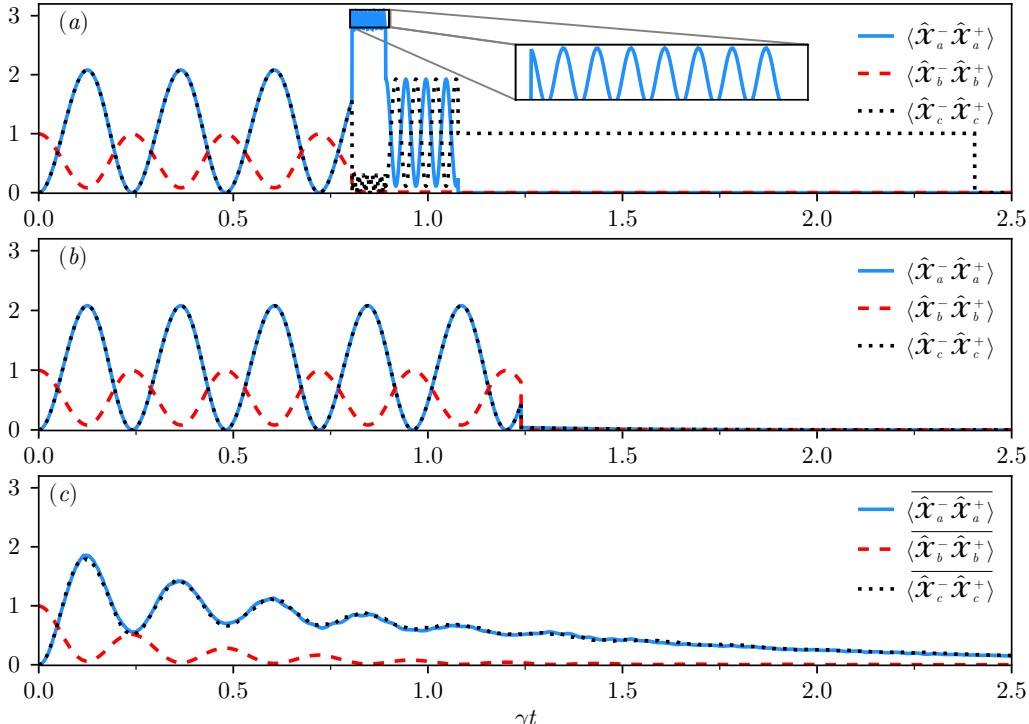

Figure 3: **Four-photon entanglement generation.** The time evolution of the mean values of the dressed number operators for different trajectories and the average over 1000 trajectories. (a) A trajectory where the first jump occurs with a photon loss from cavity $a$, projecting the dynamics into an incomplete coherent process involving the two-photon hopping between $|3, 0, 0\rangle$ and $|1, 0, 2\rangle$. A second jump puts the system into the state $|2, 0, 0\rangle$, allowing this time a complete two-photon hopping process between the two cavities [38]. When a third jump occurs with a photon loss from cavity $a$, the system jumps to the locked state $|1, 0, 0\rangle$, and the coherent dynamics is lost. After a certain time, a fourth jump leaves the system in its ground state. (b) The occurrence of a jump with a phonon loss brings the system directly to its ground state, because of the lack of energy. (c) The average behavior shows the coherent and dissipative energy exchange between the bare state $|0, 1, 0\rangle$ and the entangled state $|\psi_-^{(4e)}\rangle$. The used parameters are: $\omega_a = \omega_c \approx 0.2566\omega_b$, $g = 0.03\omega_b$, and $\gamma_a = \gamma_b = \gamma_c = 2 \times 10^{-5}\omega_b$.

in analogy to Section 2.1. The above state again encompasses entangled cavities but now with a higher number state. Compared to the two-photon entanglement, we now get a third-order process, which results in a greater sensitivity to finding the resonance point of maximum interaction. This gives a small difference between the resonance condition obtained analytically with the effective Hamiltonian and the one required from the full Hamiltonian of Eq. (1). Therefore, to find the maximum interaction point, we obtain $\omega$ through a numerical optimization process, showing a very small difference from the analytical value, but large enough to make this third-order process incomplete if using the analytical point.

As can be seen from Fig. 3(a) the trapping effect is more cumbersome. Once a measurement occurs in cavity $a$, it rips away one photon from a previously four-photon entangled state. This measurement projects the dynamics into an incomplete two-photon hopping process, between the states $|3, 0, 0\rangle$ and $|1, 0, 2\rangle$. Indeed, the specific resonance condition we choose here does not match with that involving this subprocess, and for this reason, we have an incomplete coherent dynamic. A second measurement restores the inherent parity, washing away

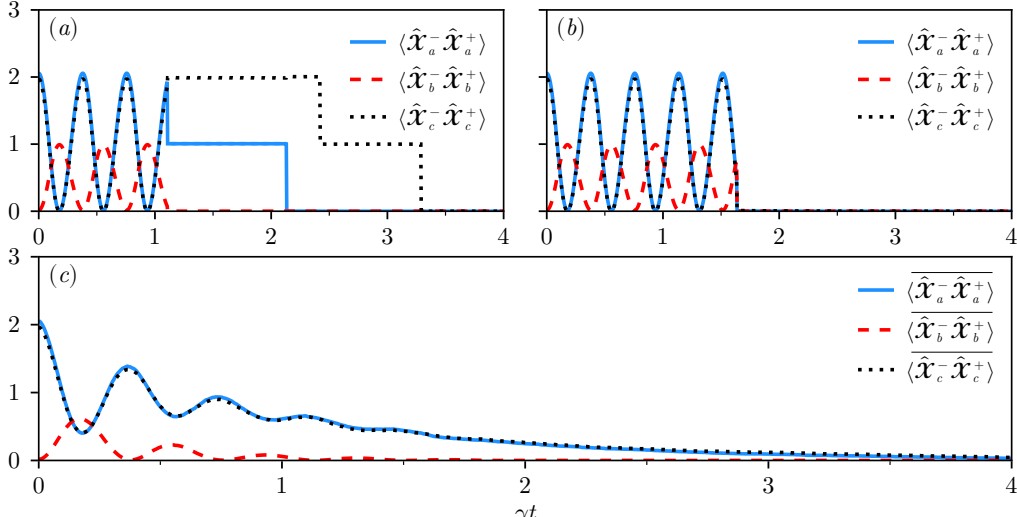

Figure 4: **The Janus effect.** Time evolution of the mean values of the dressed number operators for both cavities and the mirror. In the (a) panel a first jump causes the initial state $|2,0,2\rangle$ to collapse to the state $|1,0,2\rangle$ with consequent trapping. Indeed, from now on, there are no sub-processes involved, and the dynamics becomes trivial. In panel (b) the initial dynamics is interrupted by the measurement of the phonon, leaving the system in its ground state. Panel (c) shows the average over 1000 trajectories. The used parameters are: $\omega_a = \omega_b/4 + \varepsilon$, $\varepsilon = \omega_b/15$, $\Omega \approx 0.6067$, $g = 0.05\omega_b$, $\gamma_a = \gamma_b = \gamma_c = 2 \times 10^{-5}\omega_b$.

these spurious beatings, and letting the two cavities interact under the influence of a complete two-photon hopping interaction [38]. When a third jump occurs with a photon loss, from the cavity $a$ in our case, the system jumps to the locked state $|1,0,0\rangle$, while the coherent dynamics is lost. After a certain time, a fourth jump leaves the system in its ground state. In Fig. 3(b) we show how the occurrence of a jump with a phonon loss brings the systems directly to its ground state, because of the lack of energy. The average behavior obtained by averaging over 1000 trajectories shows the coherent and dissipative energy exchange between the bare state $|0,1,0\rangle$ and the entangled state $|\psi_-^{(4e)}\rangle$. This is reported in Fig. 3(c). Note that, the averaging process hides the complex dynamics described above. The parameters used to reproduce Fig. 3 are: $\omega_a = \omega_c \approx 0.2566\omega_b$, $g = 0.03\omega_b$, and $\gamma_a = \gamma_b = \gamma_c = 2 \times 10^{-5}\omega_b$.

## 2.3   Janus effect

We now move to an asymmetric case, $\omega_a \neq \omega_c$ (i.e., the mirror not in the middle), in which the resonance condition is expressed as $\omega_b \simeq 2(\omega_a + \omega_c)$. Under this condition the effective Hamiltonian in Eq. (A.4) up to the third order becomes

$$
\begin{aligned}
\hat{H}_{\text{eff}} =& \hat{H}_0 + \hat{H}_{\text{shift}} + \hat{H}_{\text{Jan}}, \\
\hat{H}_{\text{shift}} =& \Omega_a \hat{a}^\dagger \hat{a} + \Omega_b \hat{b}^\dagger \hat{b} + \Omega_c \hat{c}^\dagger \hat{c} + \alpha_a \hat{a}^{\dagger 2}\hat{a}^2 + \alpha_c \hat{c}^{\dagger 2}\hat{c}^2 \\
& + \alpha_{a,b} \hat{a}^\dagger \hat{a} \hat{b}^\dagger \hat{b} + \alpha_{a,c} \hat{a}^\dagger \hat{a} \hat{c}^\dagger \hat{c} + \alpha_{b,c} \hat{b}^\dagger \hat{b} \hat{c}^\dagger \hat{c}, \\
\hat{H}_{\text{Jan}} =& g_{\text{eff}} \left( \hat{a}^{\dagger 2}\hat{c}^{\dagger 2}\hat{b} + \hat{a}^2 \hat{c}^2 \hat{b}^\dagger \right),
\end{aligned}
\tag{9}
$$

where the corresponding coefficients and also the coupling strength $g_{\text{eff}}$ are written in Appendix A. Note that, $g_{\text{eff}} = 0$ if $\omega_a = \omega_c$. We call the last interaction Hamiltonian the Janus interaction because the exchange is bilateral: for each phonon, two photons are simultaneously generated in each cavity, and conversely. Now, the projecting space for the first reduced

dynamics is simply spanned by the states $\{|0,1,0\rangle, |2,0,2\rangle\}$, and the Hamiltonian takes the form

$$H = \begin{pmatrix} \Omega_b & 2g_{\text{eff}} \\ 2g_{\text{eff}} & 2(\Omega_a + \Omega_c + \alpha_a + \alpha_c + 2\alpha_{a,c}) \end{pmatrix}. \tag{10}$$

The point of maximum interaction can be found again by equating the two diagonal terms. However, as for the four-photon entanglement, this is a third-order process, and a more accurate resonance condition is found numerically. Since we need $\omega_a \neq \omega_c$, we can fix $\omega_a = \omega_b/4 + \varepsilon$ and $\omega_c$ to satisfy the required resonance condition.

Here we choose $\varepsilon = \omega_b/15$, and a numerical optimization procedure allows us to find the value of $\omega_c$ (and so also $\Omega$) to get symmetric eigenstates in terms of $|0,1,0\rangle$ and $|2,0,2\rangle$. Under this condition, the two eigenstates are

$$|\psi_{\pm}^{(\text{Jan})}\rangle = \frac{1}{\sqrt{2}}(|0,1,0\rangle \pm |2,0,2\rangle).$$

As before, we employ a quantum trajectory approach to investigate how a measurement affects the dynamics of the system. In Fig. 4(a) a single quantum jump collapses the initial state $|2,0,2\rangle$ into $|1,0,2\rangle$. The latter state has no sub-processes and exhibits trivial dynamics. Fig. 4(b) shows how the phonon measurement changes the initial dynamics and leaves the system in the ground state, in analogy to Section 2.2. Fig. 4(c) represents the average over 1000 trajectories. Note that, the "averaging" process hides the complex dynamics described above. We used the following parameters for our simulation: $\omega_a = \omega_b/4 + \varepsilon$, $\varepsilon = \omega_b/15$, $\Omega \approx 0.6067$, $g = 0.05\omega_b$, $\gamma_a = \gamma_b = \gamma_c = 2 \times 10^{-5}\omega_b$.

## 3 Conclusions

In this work, we have investigated the quantum phenomena that arise from a cavity resonator containing a two-sided perfect mirror, which acts as a mechanical oscillator interacting with two separated electromagnetic fields by radiation pressure. We have shown that, depending on the chosen resonant conditions, this system can generate $2n$-photon entanglement and bilateral photon pair emission.

We have also explored the effects of the system's parameters, such as the coupling strengths, the bare frequencies, and the initial conditions, on the entanglement and the photon emission. We have provided analytical and numerical results to support our findings and to illustrate the feasibility of observing these phenomena in realistic setups. Our work contributes to the development of quantum-enabled devices that rely on the deterministic creation and distribution of entangled states. Among them, the NOON states, emerging from the $2n$-photon entanglement, are a promising path for quantum sensing and quantum metrology. Moreover, in this work, we exploited high-order effects emerging from the standard Casimir-like interaction between mechanical objects and light fields. The Janus effect is an example, where we demonstrated the simultaneous conversion of phonons into photons in distinct modes.

We have proposed circuit-optomechanical systems and quantum simulators as possible platforms to implement our scheme, which could also be extended to other physical systems and energy scales. Furthermore, our work opens up new avenues for exploring quantum effects in tripartite systems. Our findings are expected to stimulate further research in this direction and foster the advancement of quantum technologies.

# Acknowledgments

**Funding information** F.N. is supported in part by: Nippon Telegraph and Telephone Corporation (NTT) Research, the Japan Science and Technology Agency (JST) [via the CREST Quantum Frontiers program Grant No. JPMJCR24I2,the Quantum Leap Flagship Program (Q-LEAP), and the Moonshot R&D Grant Number JPMJMS2061], and the Office of Naval Research (ONR) Global (via Grant No. N62909-23-1-2074). S.S. acknowledges the Army Research Office (ARO) (Grant No. W911NF-19-1-0065). R.L.F. acknowledges support from MUR (Ministero dell'Università e della Ricerca) through the following projects: PNRR Project ICON-Q – Partenariato Esteso NQSTI – PE00000023 – Spoke 2 – CUP: J13C22000680006, PNRR Project QUANTIP – Partenariato Esteso NQSTI – PE00000023 – Spoke 9 – CUP: E63C22002180006, PNRR Project AQuSDIT – Partenariato Esteso SERICS – PE00000014 – Spoke 5 – CUP: H73C22000880001. R.L.F. and E.R. also acknowledge support from the PNRR Project PRISM – Partenariato Esteso RESTART – PE00000001 – Spoke 4 – CUP: E13C22001870001.

# A  Effective Hamiltonian

The effective Hamiltonian, which shows in a direct way the high-order processes, can be obtained through the Schrieffer-Wolff (SW) transformation [52–54]. We start by evaluating the following rotation

$$\hat{H}_{\text{eff}} = e^{\lambda \hat{S}} \left( \hat{H}_0 + \lambda \hat{H}_I \right) e^{-\lambda \hat{S}}, \tag{A.1}$$

where

$$\hat{H}_0 = \omega_a \hat{a}^\dagger \hat{a} + \omega_b \hat{b}^\dagger \hat{b} + \omega_c \hat{c}^\dagger \hat{c}, \tag{A.2}$$

$$\hat{H}_I = \frac{g}{2} \left[ \left( \hat{a} + \hat{a}^\dagger \right)^2 - \Omega^2 \left( \hat{c} + \hat{c}^\dagger \right)^2 \right] \left( \hat{b} + \hat{b}^\dagger \right),$$

and $\lambda$ tracks how many times we apply the off-diagonal terms. Using the Baker-Campbell-Hausdorff lemma

$$e^{\hat{B}} \hat{A} e^{-\hat{B}} = \hat{A} + \left[ \hat{B}, \hat{A} \right] + \frac{1}{2} \left[ \hat{B}, \left[ \hat{B}, \hat{A} \right] \right] + \ldots + \frac{1}{n!} \underbrace{\left[ \hat{B}, \left[ \hat{B}, \left[ \hat{B}, \ldots \left[ \hat{B} \right., \hat{A} \right] \right] \right] \right]}_{n \text{ times}} + \ldots, \tag{A.3}$$

we have (up to the third order on the expansion)

$$\begin{aligned}
\hat{H}_{\text{eff}} &= \hat{H}_0 + \lambda \hat{H}_I + \lambda \left[ \hat{S}, \hat{H}_0 + \lambda \hat{H}_I \right] + \frac{\lambda^2}{2!} \left[ \hat{S}, \left[ \hat{S}, \hat{H}_0 + \lambda \hat{H}_I \right] \right] \\
&\quad + \frac{\lambda^3}{3!} \left[ \hat{S}, \left[ S, \left[ \hat{S}, \hat{H}_0 + \lambda \hat{H}_I \right] \right] \right] + \mathcal{O}(\lambda^4) \\
&= \hat{H}_0 + \lambda \left( \hat{H}_I + \left[ \hat{S}, \hat{H}_0 \right] \right) + \frac{\lambda^2}{2!} \left( 2! \left[ \hat{S}, \hat{H}_I \right] + \left[ \hat{S}, \left[ \hat{S}, \hat{H}_0 \right] \right] \right) \\
&\quad + \frac{\lambda^3}{3!} \left( \frac{3!}{2!} \left[ \hat{S}, \left[ \hat{S}, \hat{H}_I \right] \right] + \left[ \hat{S}, \left[ \hat{S}, \left[ \hat{S}, \hat{H}_0 \right] \right] \right] \right) + \mathcal{O}(\lambda^4).
\end{aligned}$$

In order to cancel the linear term in $\lambda$, we now choose $\hat{S}$ such that $[\hat{S}, \hat{H}_0] = -\hat{H}_I$, and the total effective Hamiltonian becomes

$$\begin{aligned}
\hat{H}_{\text{eff}} &\simeq \hat{H}_0 + \frac{\lambda^2}{2} \left[ \hat{S}, \hat{H}_I \right] + \frac{\lambda^3}{3} \left[ \hat{S}, \left[ \hat{S}, \hat{H}_I \right] \right] \\
&= \hat{H}_0 + \lambda^2 \hat{H}_{\text{eff}}^{(2)} + \lambda^3 \hat{H}_{\text{eff}}^{(3)}.
\end{aligned} \tag{A.4}$$

The most crucial step in doing SW transformation is to get the generator $\hat{S}$, such that $[\hat{S}, \hat{H}_0] = -\hat{H}_I$. In the following, we apply a systematic method to obtain it. We impose $\hat{S} = [\hat{H}_0, \hat{H}_I]$, and, leaving the coefficients undefined since they will be obtained using the condition $[\hat{S}, \hat{H}_0] = -\hat{H}_I$, we get

$$
\begin{aligned}
\hat{S} = \left[\hat{H}_0, \hat{H}_I\right] = {} & c_0 \hat{b} + c_1 \hat{b}^\dagger + c_3 \hat{c}^{\dagger 2} \hat{b} + c_4 \hat{a}^\dagger \hat{b}^\dagger \hat{a} \\
& + c_5 \hat{b}^\dagger \hat{c}^\dagger \hat{c} + c_6 \hat{c}^\dagger \hat{b} \hat{c} + c_7 \hat{a}^{\dagger 2} \hat{b} + c_8 \hat{a}^2 \hat{b} + c_9 \hat{a}^{\dagger 2} \hat{b}^\dagger \\
& + c_{10} \hat{b}^\dagger \hat{a}^2 + c_{11} \hat{a}^\dagger \hat{a} \hat{b} + c_{12} \hat{b}^\dagger \hat{c}^2 + c_{13} \hat{b}^\dagger \hat{c}^{\dagger 2} + c_{14} \hat{b} \hat{c}^2 .
\end{aligned}
$$

By using $[\hat{S}, \hat{H}_0] = -\hat{H}_I$, the generator becomes

$$
\begin{aligned}
\hat{S} = {} & \frac{g(1-\Omega^2)}{2\omega_b} \left(\hat{b} - \hat{b}^\dagger\right) + \frac{g}{\omega_b} \hat{a}^\dagger \hat{a} \left(\hat{b} - \hat{b}^\dagger\right) - \frac{\Omega^2 g}{\omega_b} \hat{c}^\dagger \hat{c} \left(\hat{b} - \hat{b}^\dagger\right) - \frac{g}{4\omega_a - 2\omega_b} \left(\hat{a}^{\dagger 2} \hat{b} - \hat{a}^2 \hat{b}^\dagger\right) \\
& + \frac{g}{4\omega_a + 2\omega_b} \left(\hat{a}^2 \hat{b} - \hat{a}^{\dagger 2} \hat{b}^\dagger\right) + \frac{\Omega^2 g}{4\omega_c - 2\omega_b} \left(\hat{c}^{\dagger 2} \hat{b} - \hat{c}^2 \hat{b}^\dagger\right) \\
& - \frac{\Omega^2 g}{4\omega_c + 2\omega_b} \left(\hat{c}^2 \hat{b} - \hat{c}^{\dagger 2} \hat{b}^\dagger\right) ,
\end{aligned}
\tag{A.5}
$$

and the perturbative Hamiltonians $\hat{H}_{\text{eff}}^{(2)}$ and $\hat{H}_{\text{eff}}^{(3)}$ for the second and third order respectively can be obtained following Eq. (A.4).

The total effective Hamiltonian expressed in Eq. (A.4) describes all the high-order processes up to the third order. By imposing a specific resonance condition, one can make a process dominant over the others. By applying the rotating wave approximation (RWA) to $\hat{H}_{\text{eff}}$, we reduce it to a specific effective one that describes that specific process. As done in the main text, we want to explore the generation of $2n$-photon entanglement (e.g., two-, four-photon) and the bilateral photon emission, namely, the Janus effect. In particular, the two-photon entanglement is obtained by choosing $\omega_a \approx \omega_b \approx \omega_c$, leading to different oscillating terms, which can be neglected by applying the RWA. All the remaining non-oscillating terms form the specific effective Hamiltonian, expressed in Eq. (3). Similarly, the same procedure can be performed in the case of the four-photon entanglement ($\omega_b \approx 4\omega$, with $\omega = \omega_a = \omega_c$), leading to the specific effective Hamiltonian in Eq. (6), and in the case of the Janus effect ($\omega_b \approx 2(\omega_a + \omega_c)$) in Eq. (9). In the case of the Janus effect, here we show all the coefficients contained inside Eq. (9):

$$
\left\Vert
\begin{array}{cc}
\Omega_a = \frac{g^2(\Omega^3 + 2\Omega^2 - 3\Omega - 5)}{\omega_b(\Omega + 2)} & \Omega_b = \frac{g^2(\Omega^8 + 3\Omega^7 + 2\Omega^6 + 2\Omega^2 + 3\Omega + 1)}{\Omega \omega_b(2\Omega^2 + 5\Omega + 2)} \\[2ex]
\Omega_c = \frac{\Omega^2 g^2(-5\Omega^3 - 3\Omega^2 + 2\Omega + 1)}{\omega_b(2\Omega + 1)} & \alpha_a = \frac{g^2(-3\Omega^2 - 6\Omega - 1)}{2\Omega \omega_b(\Omega + 2)} \\[2ex]
\alpha_c = \frac{\Omega^4 g^2(-\Omega^2 - 6\Omega - 3)}{2\omega_b(2\Omega + 1)} & \alpha_{a,b} = \frac{2g^2(\Omega + 1)}{\Omega \omega_b(\Omega + 2)} \\[2ex]
\alpha_{a,c} = \frac{2\Omega^2 g^2}{\omega_b} & \alpha_{b,c} = \frac{2\Omega^5 g^2(\Omega + 1)}{\omega_b(2\Omega + 1)}
\end{array}
\right\Vert
$$

while the the effective coupling is $g_{\text{eff}} = \frac{\Omega g^3(2\Omega^6 + 5\Omega^5 + 4\Omega^4 - 4\Omega^2 - 5\Omega - 2)}{2\omega_b^2 \cdot (2\Omega^2 + 5\Omega + 2)}$. After the RWA, the result is equal to that obtained with other procedures, such as the generalized James' effective Hamiltonian method [65].

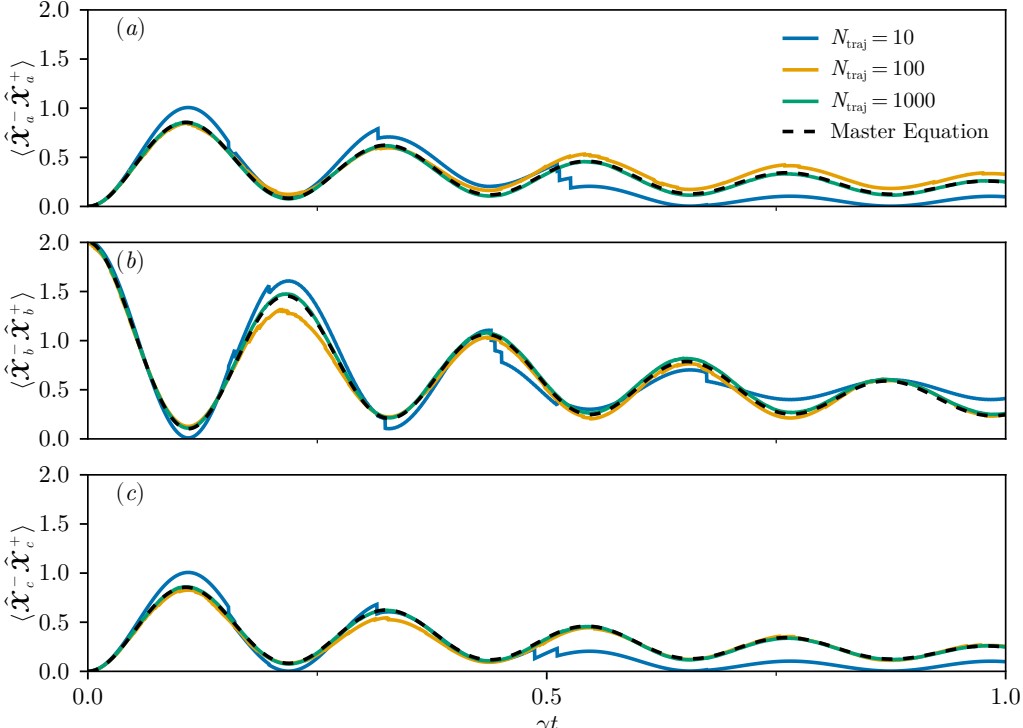

Figure 5: Comparison of the time evolution between different numbers of quantum trajectories and the convergence to the master equation case, for the left cavity (a), mirror (b), and right cavity (c) mean excitation number. A clear convergence behavior can be seen when increasing the number of trajectories to average.

## B Convergence of the quantum trajectories to the master equation

The behavior described by the Lindblad master equation is recovered when averaging over a large number of quantum trajectories [66–69]. Here, we show the convergence of the quantum trajectories by varying the number of trajectories.

The time evolution of the density matrix is governed by the following master equation

$$\dot{\rho} = -i[\hat{H}, \hat{\rho}] + \sum_{o \in \{a,b,c\}} \gamma_o \mathcal{D}[\hat{\mathcal{X}}_o^+]\hat{\rho}, \tag{B.1}$$

where

$$\mathcal{D}[\hat{O}]\hat{\rho} = \hat{O}\hat{\rho}\hat{O}^\dagger - \frac{1}{2}\hat{O}^\dagger\hat{O}\hat{\rho} - \frac{1}{2}\hat{\rho}\hat{O}^\dagger\hat{O} \tag{B.2}$$

is the Lindblad dissipator. This approach implicitly averages the outcomes through the density matrix description.

An equivalent approach can be formulated in terms of quantum trajectories, where the system evolves through the effective Hamiltonian

$$\hat{H}_{\text{eff}}^{(\text{traj})} = \hat{H} - \frac{i}{2}\sum_{o \in \{a,b,c\}} \gamma_o \hat{\mathcal{X}}_o^- \hat{\mathcal{X}}_o^+. \tag{B.3}$$

This is a non-unitary dynamics, and the norm of the evolving state is no longer conserved ($\langle\psi(0)|\psi(0)\rangle = 1$, $\langle\psi(t)|\psi(t)\rangle \neq 1$). A quantum jump occurs when the norm drops below a given random number $r$. After a jump at $t = t_1$, the new state is given by

$$|\psi(t_1 + \delta t)\rangle = \frac{\hat{\mathcal{X}}_o^+ |\psi(t_1)\rangle}{\sqrt{\langle\psi(t_1)|\hat{\mathcal{X}}_o^- \hat{\mathcal{X}}_o^+|\psi(t_1)\rangle}}. \tag{B.4}$$

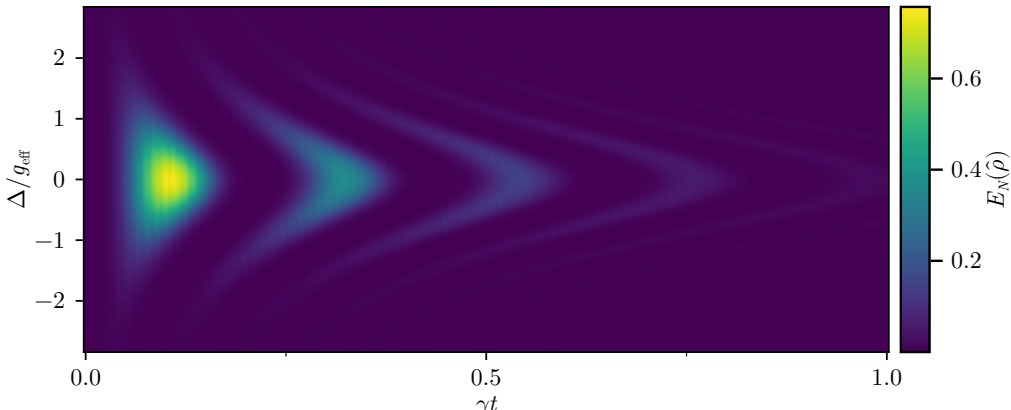

Figure 6: Evolution of the logarithmic negativity entanglement as a function of time and detuning $\Delta$. The efficiency of the entanglement generation decreases as the detuning increases.

Fig. 5 shows the convergence of the quantum trajectories to the master equation, when increasing the number of trajectories.

## C    Entanglement behavior on detuned resonances

The effective Hamiltonians derived in this work emerged from specific resonance conditions. For instance, the two-photon entangled state generation is activated when $\omega_a \simeq \omega_b \simeq \omega_c$. A good question one may ask is whether this effect is still visible when the resonances don't match exactly. Here we discuss this topic, applying it to the two-photons entangled state generation case, without loss of generality for the other cases.

The effective Hamiltonian in Eq. (3) gives an interaction between the states $|0,2,0\rangle$ and $|\psi_+^{(2e)}\rangle$, with the effective coupling $g_{\text{eff}} = 2\sqrt{2}g^2/\omega_b$, derived from the off-diagonal elements of the matrix in Eq. (4). This results in Rabi oscillations when initializing the system in one of the two states, with a coherent exchange of energy between the mechanical and the photonic subparts. The dressed resonance condition can be derived by equating the diagonal terms in the matrix of Eq. (4), which gives $\omega_{\text{res}} \equiv \omega_b + g^2/\omega_b$ as the frequency that the cavities $a$ and $c$ need to match the resonance. By defining $\Delta \equiv \omega_b - \omega_{\text{res}}$ (with $\omega_c = \omega_a$), the time evolution can be derived analytically. Indeed, the population of the two cavities oscillates at the detuning-dependent Rabi frequency $\Omega(\Delta) = \sqrt{g_{\text{eff}}^2 + (\Delta/2)^2}$, and its amplitude scales as $\mathcal{A}(\Delta) = g_{\text{eff}}^2/(g_{\text{eff}}^2 + (\Delta/2)^2)$ [70]. This results in a decrease in the visibility of the generation of the $|\psi_+^{(2e)}\rangle$ state as the detuning increases. In general, $g_{\text{eff}}^2/\Delta \ll 1$ is a good condition for achieving a good efficiency of the process.

Fig. 6 shows the logarithmic negativity entanglement as a function of time and detuning. The entanglement is obtained by calculating the logarithmic negativity

$$E_N(\hat{\rho}) = \log_2 ||\hat{\rho}^{\Gamma_A}||_1\,, \tag{C.1}$$

where $\hat{\rho}^{\Gamma_A}$ is the partial transpose of $\hat{\rho}$ with respect to the subsystem $A$, and $||\cdot||_1$ is the trace norm. Since we are interested in the two-photon entanglement, we first trace out the mechanical part, and we perform the partial transpose with respect to the cavity $a$. As can be seen from the Chevron pattern in Fig. 6, the efficiency of the entanglement generation decreases as the cavity-phonon detuning increases.

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
