# Peer review of "Bilateral photon emission from a vibrating mirror and multiphoton entanglement generation"

_SciPost Physics, doi:SciPost Phys. 18, 067 (2025)_

## Round 2 · Referee Report · Anonymous (Referee 2) · 2024-12-17

Strengths

1-Rich physics described in the manuscript
2-Analytical results well documented
3-Numerical analysis supports results well
4-References are appropriate and comprehensive
5-Clear description of the scheme and initialisation
6-Discussion of convergence clear
7-Disorder-tolerance evaluated and a clear figure of merit is given

Weaknesses

1-Clarification of entanglement in the system correct but too brief
2-Figures "showing" convergence not helpful
3-Experimental realisation remains speculative

Report

In the updated manuscript on "Bilateral photon emission from a vibrating mirror and multiphoton entanglement generation" the authors have clearly improved the clarity of their already interesting results even though there remain few minor comments that need to be addressed.

Reply to comment 2:
It is worth distinguishing between the system initialization and the state-conversion process (i.e., 2n-photon entangled state generation). Obviously, one needs a drive to initialize the system, otherwise we simply stay in the ground state. On the other hand, after the system is initialized, we can switch off the drive (or a pulse sequence), and leave the spontaneous generation of the entangled state due to the Casimir-Rabi oscillations[^6] derived in our manuscript.

As an example of the system initialization, in the case of Fig. (2) our the current manuscript, the |0,2,0⟩ state can be obtained by using several procedures (see for example Fig 3 of Ref. [^1], or other references [^2][^3][^4][^5]). Nonetheless, we thank the Referee for raising this question and have added a comment in the updated manuscript.

Response to reply to comment 2:
I think that the comment in the updated manuscript clearly addresses the necessary distinction between the state initialisation and the following conversion. This should suffice for the ordinary audience (such as myself) to understand that there is a prior initialisation before the physics under investigation take place.

Reply to comment 3:
We take the opportunity to address this second concern raised by the Referee.
We have to distinguish between the 2n-photon entangled state $|ψ^{(2e)}_+⟩=\frac{1}{\sqrt{2}}(|2,0,0⟩+|0,0,2⟩)$ (which is deterministically generated after half-period of the Rabi oscillation: $t=\frac{π}{2g_{\text{eff}}}$, where $g_{\text{eff}}=2\sqrt{2}g^2/ω_b$), and the entangled state $|ψ(t)⟩=c_1(t)|0,2,0⟩+c_2(t)|ψ^{(2e)}_+⟩$ which is governed by the Rabi oscillation itself. Although in our manuscript we mainly focus on the $|ψ^{(2e)}_+⟩$ entangled state, we have a multipartite entangled state between the mirror and the two photonic fields. Nonetheless, we see the photonic correlations even during the Rabi oscillation.
Since the state $|ψ^{(2e)}_+⟩$ has a straightforward analytical form, we decided not to explicitly quantify the entanglement entropy of this state. Indeed, it is straightforward to see that it is a non-factorizable pure state. This state belongs to the class of the NOON states (which are very important in quantum sensing and metrology), and their entanglement is very well quantified (see, e.g., Ref. [^7]).
Additionally, in response to Referee’s Comment 5 (see below), we have added an Appendix that presents the logarithmic negativity entanglement as a function of the subsystems’ detuning.

Response to reply to comment 3:
While I entirely agree with all the facts in the statement, my initial reason to address this issue is the discussion of Fig. 2 in the main text. I agree that "when the cavity $c$ jumps, the number of photons in the cavity $a$ immediately goes to zero, as a clear signature of the quantum correlations exhibited in the entangled state $|{\psi^{(2e)}_{i}}$>".
But then applying the same logic to Fig. 2(a) should result in a clear indication of quantum correlations when "the first jump occurs with a phonon loss, [the system is locked] to the state $|0,1,0>$ ", i.e. the photons in both cavities immediately go to zero. As you said, the state is a multipartite entangled state. Therefore, I think that the photons in both cavities immediately going to zero is a signature of the quantum correlations of the multipartite entangled state which should be clearly stated in the discussion of Fig. 2(a). The text does not mention anything about the quantum correlations in this case. Therefore, it can be misinterpreted$-$as I initially did$-$as a statement that there are no quantum correlations in Fig. 2(a) which (I think) we all agree is wrong.

Reply to comment 4:
The mathematical equivalence between the Lindblad master equation and quantum trajectory methods, such as the Monte Carlo wave function (MCWF) method, is well established in the literature (see, e.g., Refs. [^8][^9][^10][^11]). Nonetheless, we have added an Appendix in the revised manuscript showing the convergence behavior of quantum trajectories toward the master equation as a function of the number of trajectories.

Response to reply to comment 4:
Again I have nothing to complain about regarding the facts. I would only remark that Fig. 5 in its current form does not help in seeing the convergence behaviour of the MCWF method to the Lindblad master equation. I believe Fig. 5 would be largely improved if it consisted of three panels where each panel shows only one the physical quantities for the different number of realisations and the master equation. This should show the convergence of the numerical method to the Lindblad master equation for all quantities under investigation.

Reply to comment 5:
The effects studied in this manuscript rely on specific resonance conditions between the three subsystems. For example, the generation of the 2-photon entangled state can be achieved under the condition $ω_a≃ω_b≃ω_c$. The interaction term enabling this dynamic is given by
$\hat{H}_{\text{ent}}=−\frac{g^2}{ω_b}[(\hat{a}^2+\hat{c}^2)\hat{b}^{\dagger 2}+\text{h.c.}]$
in Eq. (3) of the current manuscript. The efficiency of the conversion process between the states $|0,2,0⟩$ and $|ψ^{(2e)}_+⟩$ depends on the ratio $|g_{\text{eff}}|/|ω_{a,c}−ω_b|$, where $g_{\text{eff}}=2\sqrt{2}g^2/ω_b$ (see the off-diagonal elements in the matrix in Eq.(4) of the current manuscript). Thus, it depends not only on the effective coupling but also on the detuning. Notably, the efficiency varies continuously with detuning, allowing for some relaxation of the resonance conditions.
We have added a new section in the Appendix that examines the behavior of this process as a function of detuning.

Response to reply to comment 5:
The new Appendix and the analysis in Fig. 6 is very helpful in understanding the influence of the most relevant disorder, namely in the mechanical frequency $\omega_b$. I believe that $\omega_a=\omega_c$ should generally be achievable and is thus of lesser interest. However, I would like to know if the authors considered disorder in this resonance condition and have some insights when $\omega_a \ne \omega_c$.

Reply to comment 6:
As already mentioned in the manuscript, we are aware of the current experimental challenges in constructing such setups and agree with the Referee’s concerns. Nonetheless, we would like to point out that:
1. The quadratic interaction term presented in the manuscript is intrinsic in the optomechanical interaction, and it is already derived in several works (see, e.g., [^12] and its circuit analog [^13]).
2. The state-of-art technology in optomechanics is advancing rapidly, and we recognize the potential for achieving this regime in the future.
On point 2, we would like to discuss the current state of optomechanics technology.
Over the past two decades, circuit optomechanics has achieved remarkable results. Cryogenic cooling can bring a microwave-frequency mechanical mode (4–6 GHz) to its quantum ground state [^14]. Additionally, experiments have demonstrated resonant quantum interactions between a superconducting phase qubit and mechanical modes, modeled by the quantum Rabi Hamiltonian [^15]. The addition of a quantum two-level system has increased coupling strength and non-linearities, bringing the radiation-pressure interaction near the strong-coupling regime. [^16]. This is achieved with a Josephson junction qubit setup operating in the microwave regime. By analyzing the coupling between a mechanical resonator and a flux qubit, and using the experimentally achieved qubit-oscillator coupling strength parameter in Ref.[^14], the Supplemental Material of Ref.[^17] shows that the radiation-pressure interaction strength between the high-frequency mechanical resonator and an electromagnetic one can be achieved with this technology.
Given advances in circuit-QED, it is likely that our proposed results could be observed on this platform.
Despite these advancements, circuit-QED analogs of optomechanics may provide an additional suitable platform [^18][^19]. For instance, the mechanical membrane can be simulated by a Superconducting Quantum Interference Device (SQUID). As an example, it's worth to mention the demonstration of photons-pair extraction through vacuum perturbation (dynamical Casimir effect[^20][^21]).
Finally, as stated in the manuscript, we hope that this theoretical proposal can stimulate future experimental realizations.

Response to reply to comment 6:
I completely agree with the authors in their assessment of experimental platforms that may realise the suggested physics. Moreover, I join the authors in their hope that this theoretical proposal can stimulate future experimental realisations.

All in all, I would like to congratulate the authors on their insightful and stimulating manuscript and hope that our discussion has improved its quality. I believe that my remaining requests are minor and can be answered and implemented quickly. If these queries are responded to in a satisfactorily manner, I can recommend this manuscript for publication with a clear conscience.

Requested changes

1-I still ask for a clarification in the discussion of Fig. 2(a) or a clarification why the photons in both cavities immediately going to zero is not a signature of the quantum correlations in the multipartite entangled state. 2- I would ask Fig. 5 to be replotted such that it consists of three panels where each panel shows only one the physical quantities for $N_{\text{traj}}=10$, $N_{\text{traj}}=100$, $N_{\text{traj}}=1000$, and the Lindblad master equation to improve its clarity with regards to the convergence of the applied method. 3(optional)-I would suggest to add some comments on the impact of disorder in the optical frequencies, i.e. $\omega_a \ne \omega_c$

Recommendation

Ask for minor revision

  • validity: top
  • significance: good
  • originality: high
  • clarity: high
  • formatting: excellent
  • grammar: perfect

Author:  Alberto Mercurio  on 2025-01-19  [id 5136]

(in reply to Report 1 on 2024-12-17)

We sincerely thank the Referee for finding our manuscript stimulating. We are confident that the revisions made in this regard have significantly enhanced the manuscript’s quality.

We agree with the entire Referee's report. Below, we address the Referee’s final requests.

Referee's Requested Changes

  1. I still ask for a clarification in the discussion of Fig. 2(a) or a clarification why the photons in both cavities immediately going to zero is not a signature of the quantum correlations in the multipartite entangled state.
  2. I would ask Fig. 5 to be replotted such that it consists of three panels where each panel shows only one the physical quantities for $N_{traj}=10$$N_{traj}=100$$N_{traj}=1000$, and the Lindblad master equation to improve its clarity with regards to the convergence of the applied method. 
  3. (optional)-I would suggest to add some comments on the impact of disorder in the optical frequencies, i.e. $\omega_a \neq \omega_c$.

Author's Reply

  1. We have addressed more in detail this request, providing a discussion on the multipartite entanglement arising during the time evolution as a result of the Rabi oscillations.
  2. We have updated the figure to display the convergence of the quantum trajectories separately for each observable. We hope this revision enhances the clarity of the figure.
  3. We thank the Referee for this suggestion. Anyway, we believe that this study goes beyond the scope of the current work, which already covers the main physics we aimed to explore. This topic can be addressed in future studies, investigating the different behaviors across various frequency detuning ranges.

---

## Round 2 · List of Changes

- Included a comment in the manuscript addressing the system initialization.
- Added a section in the Appendix discussing the convergence of quantum trajectories.
- Conducted and included a study in the Appendix on energy exchange as a function of the detuning between subsystems.
- Updated the acknowledgements and affiliations.
- Provided clarification regarding the question of multipartite entanglement.

---

## Round 3 · List of Changes

- Added a comment on multipartite entanglement
- Changed the layout of the convergence criteria of quantum trajectories

---

## Editorial Decision

published